# Senescence Markers in Peripheral Blood Mononuclear Cells in Amnestic Mild Cognitive Impairment and Alzheimer’s Disease

**DOI:** 10.3390/ijms23169387

**Published:** 2022-08-20

**Authors:** Felipe Salech, Carol D. SanMartín, Jorge Concha-Cerda, Esteban Romero-Hernández, Daniela P. Ponce, Gianella Liabeuf, Nicole K. Rogers, Paola Murgas, Bárbara Bruna, Jamileth More, María I. Behrens

**Affiliations:** 1Centro de Investigación Clínica Avanzada (CICA), Facultad de Medicina-Hospital Clínico, Universidad de Chile, Santiago 8380453, Chile; 2Departamento de Neurología y Neurocirugía, Hospital Clínico Universidad de Chile, Santiago 8380430, Chile; 3Programa de Fisiología y Biofísica, Instituto de Ciencias Biomédicas (ICBM), Facultad de Medicina, Universidad de Chile, Santiago 8380453, Chile; 4Laboratorio de Obesidad y Metabolismo Energético (OMEGA), Instituto de Nutrición y Tecnología de los Alimentos (INTA), Universidad de Chile, Santiago 7830490, Chile; 5Departamento de Neurociencia, Facultad de Medicina de la Universidad de Chile, Santiago 8380453, Chile; 6Instituto de Bioquímica y Microbiología, Facultad de Ciencias, Universidad Austral de Chile, Valdivia 5110566, Chile; 7Departamento de Neurología y Psiquiatría, Clínica Alemana-Universidad del Desarrollo, Santiago 8370065, Chile

**Keywords:** cellular senescence, peripheral blood mononuclear cells, Alzheimer’s Disease, aMCI, aging

## Abstract

Recent studies suggest that cellular senescence plays a role in Alzheimer’s Disease (AD) pathogenesis. We hypothesize that cellular senescence markers might be tracked in the peripheral tissues of AD patients. Senescence hallmarks, including altered metabolism, cell-cycle arrest, DNA damage response (DDR) and senescence secretory associated phenotype (SASP), were measured in peripheral blood mononuclear cells (PBMCs) of healthy controls (HC), amnestic mild cognitive impairment (aMCI) and AD patients. Senescence-associated βeta-galactosidase (SA-β-Gal) activity, G0-G1 phase cell-cycle arrest, p16 and p53 were analyzed by flow cytometry, while IL-6 and IL-8 mRNA were analyzed by qPCR, and phosphorylated H2A histone family member X (γH2AX) was analyzed by immunofluorescence. Senescent cells in the brain tissue were determined with lipofuscin staining. An increase in the number of senescent cells was observed in the frontal cortex and hippocampus of advanced AD patients. PBMCs of aMCI patients, but not in AD, showed increased SA-β-Gal compared with HCs. aMCI PBMCs also had increased IL-6 and IL8 mRNA expression and number of cells arrested at G0-G1, which were absent in AD. Instead, AD PBMCs had significantly increased p16 and p53 expression and decreased γH2Ax activity compared with HC. This study reports that several markers of cellular senescence can be measured in PBMCs of aMCI and AD patients.

## 1. Introduction

Alzheimer’s Disease is a neurodegenerative disease and the most frequent cause of dementia [1]. Despite enormous efforts in research of beneficial pharmacological treatments for AD [2], there is still an urgent need for new therapeutic options. The lack of understanding of early disease mechanisms prevents the development of new therapeutic options. Most of the research has been focused on strategies to avoid the formation or deposition of amyloid and tau aggregates in the brain. However, AD is a complex and multifactorial disorder; therefore, alternative approaches to treat this devastating disease are peremptory.

Recent studies have demonstrated that cellular senescence, a condition in which cells undergo a stable growth arrest in response to several insults, such as DNA damage, oxidative stress and oncogenic stressors, may play a role in AD pathogenesis [2,3]. Cellular senescence is important during development and wound healing and in preventing tumor development [3]. However, during aging, senescent cells accumulate and secrete several proinflammatory and matrix-degrading molecules forming the SASP, which is associated with age-related tissue inflammation and disease [4,5,6]. Remarkably, there is evidence that the elimination of senescent cells has a positive effect on extending lifespan and improving age-related conditions [7,8].

Recently, a role for senescence in neurodegeneration has been revealed. Senescence has been reported in several cells in the brain of AD patients, such as neurons [9,10], microglia [11], progenitor oligodendrocytes [12] and neural stem cells [13]. Interestingly, preclinical models have shown that the elimination of senescent cells with the senolytics Quercetin and Dasatinib is associated with reductions in amyloid plaques and proinflammatory cytokines in the brains of the animals, along with improvements in learning and memory [12,14,15].

The identification of senescent cells is based on recognizing the hallmarks of senescence that include cell-cycle arrest, macromolecular damage response, dysregulation of cellular metabolism and the secretory phenotype [5]. Currently, there is no single gold standard marker that defines senescence by itself, so different strategies need to be applied to determine the state of senescence in cells and tissues [16]. Some well-accepted markers of senescence include the presence of SA-β-Gal, markers of proliferation arrest, increased expression of cyclin-dependent kinase (CDK) inhibitor p16 (p16^INK4A^), DDR, γH2AX and elevated SASP components, among others. Furthermore, senescent cells are heterogeneous, showing phenotypes that depend on the stressor, cell type, tissue and context [16]. Considering this wide variety of characteristics associated with cellular senescence, the use of “multi-markers workflow” is promoted by experts in the field for the identification of senescent cells [5].

AD patients have several systemic manifestations that accompany the central nervous system dysfunction [17,18]. Aβ deposits have been detected in the periphery; however, there is still controversy about whether they are released from the brain through the blood–brain barrier or generated in peripheral organs [18]. Nevertheless, this evidence suggests that AD manifestations extend beyond the brain and that AD pathogenesis is closely associated with systemic abnormalities [18,19,20,21,22]. Several studies have accumulated, showing that there is an active crosstalk between the brain and peripheral tissues, and multiple pathways altered in the brain are reflected in peripheral cells and plasma [23]. From this point of view, it is interesting to evaluate AD-associated changes in the central nervous system as well as in the periphery in amnestic mild cognitive impairment (aMCI) and AD patients using non-invasive techniques. The analysis of PBMCs has been used as a model for the study of AD; mitochondrial dysfunction, elevated apoptosis, deregulation of antioxidant response, and genetic differences have been detected in PBMCs of MCI and AD patients [19,20,21,24]. In this line, we have previously shown that PBMCs from AD patients have increased susceptibility to hydrogen peroxide (H_2_O_2_)-induced death [25], which depends on dementia severity [26,27].

Considering all this information, we hypothesize that it is possible to find cellular senescence marks in peripheral tissues of patients with AD, and the objective of this study is to search for them in PBMCs of patients at early and late stages of the disease. We show here that several hallmarks of cellular senescence, such as alterations in cell metabolism, cell-cycle arrest, DDR response and SASP, can be measured differentially in PBMCs of aMCI and AD patients.

## 2. Results

### 2.1. Senescent Mark in Frontal and Hippocampal Cells of AD Patients

As a first step, we determined the presence of senescent cells in the brain tissue of AD patients and HC by staining lipofuscin with the biotinylated Sudan Black B based chemical reagent SenTraGor, which has been validated as a marker of senescence in this type of tissue [28]. In the frontal brain tissue of CDR-3 AD patients (Braak stage V-VI), the number of senescent cells was significantly higher compared with controls (Figure 1A,C). Taking into consideration the role of the hippocampus in the pathophysiology of AD, the senescence marker was measured in the hippocampal tissue from AD patients at different stages of the disease. As in the frontal brain tissue, a significant increase in lipofuscin marker was observed in the hippocampus at Braak stages IV-V compared with HC (Figure 1B,D). There was a tendency of a greater number of senescent cells in hippocampal sections at Braak stages II-III, suggesting that senescence depends on the severity of the disease (Figure 1B,D). Taken together, these results support previous studies showing an increase in the number of senescent cells in different areas of the brain in AD patients.

### 2.2. Senescent Mark in Peripheral Blood Mononuclear Cells

Considering studies that looked for markers of neurodegenerative diseases in peripheral tissues, we explored the presence of senescent markers in peripheral blood cells of patients with aMCI and AD compared with HC, using a multi-marker workflow targeting senescence hallmarks as the framework [5]. The demographic and clinical characteristics of participants are shown in Table 1. 

As a first step, senescence screening was measured in PBMCs by cytometric determinations of SA-β-Gal activity using its fluorescent substrate C12FDG. An increase in lysosomal SA-β-Gal is one of the most widely used methods to identify senescent cells as part of their deregulated metabolism [5,29]. A significant increase in the C12FDG mark was observed in aMCI patients compared with HC (Figure 2A). Interestingly, this increase in senescence was not observed in PBMCs from AD patients who showed senescence levels comparable to HC.

Growth arrest is another key feature of senescent cells. We determined the percentage of PBMCs cells arrested at the G0-G1 phase of the cell cycle. Our results showed an increase in the percentage of PBMCs arrested at G0-G1 in aMCI patients compared with HC (Figure 2B). The PBMCs of AD patients showed G0-G1 percentage not significantly different from the controls or aMCI.

Cell-cycle blockade in senescent cells converged on the p53–Cdkn1a (p21) and/or retinoblastoma (RB)–Cdkn2a (p16) pathways. We determined the expression of p16 and p53 in PBMCs of AD and aMCI patients. There was a significant increase in the percentage of cells that express p16 and p53 in AD patients compared with HC; a non-significant increase in p53 was observed in aMCI patients (Figure 3A,B).

The activation of the DNA damage response (DDR) in senescence cells leads to the activation of p53 and the phosphorylation of the DDR orchestrator histone, γH2AX. The immunofluorescence determination of γH2AX showed significantly reduced levels in AD PBMCs compared with both healthy controls and aMCI patients (Figure 4).

We then looked for the mRNA synthesis of cytokines IL-6 and IL-8 by PBMCs, as both are part of the senescence-associated secretory phenotype known as SASP. aMCI patients showed an increased expression of IL-6 and IL8, which was not present in AD patients (Figure 5A,B). This suggests that the mRNA levels of IL-6 and IL-8 might vary along the course of the disease, with increased levels at early stages, which decrease as the disease progresses (Figure 5A,B).

Finally, the plasmatic levels of IL-6 and IL-8 proteins were measured with ELISA (Figure 5C,D). There were no significant differences in the levels of IL-6 in aMCI patients vs. HC, with a trend toward increased IL-6 in AD, but the low number of patients and the dispersion of the data do not allow for a conclusive determination (Figure 5C). Instead, the plasmatic levels of IL-8 were significantly higher in aMCI patients than in healthy controls but down to control levels in AD patients (Figure 5D).

## 3. Discussion

To our knowledge, this is the first study to report changes in markers associated with cellular senescence in PBMC of patients at different stages of AD, starting at the aMCI stage. Our results show the presence of several hallmarks of senescence, including alterations in cell metabolism, cell-cycle arrest, DDR response and SASP. These markers, associated with the increase in lipofuscin signal in the brain tissue of AD patients, found by other authors and us, open up the possibility of having a peripheral tissue sensor for this phenomenon.

Geroscience is the interdisciplinary field that aims to understand the relationship between aging and age-related diseases under the hypothesis that chronic disease and aging share common molecular mechanisms, generating a novel framework for searching for new therapeutic targets [30]. Several hallmarks of aging, including genomic instability, telomere attrition, epigenetic alterations, loss of proteostasis, mitochondrial dysfunction, cellular senescence, deregulated nutrient sensing, stem cell exhaustion and altered intercellular communication, participate in the pathophysiology of neurodegenerative diseases [31]. We show here that one of the hallmarks of aging, senescence, is present in the brain and PBMCs of patients with an age-associated chronic disease, such as AD.

Cellular senescence was initially thought to be important for controlling malignancy, but now, senescence is known to participate in other important cell functions, such as tissue remodeling in wound repair and embryogenesis [32,33]. Furthermore, its association with many aging and age-related diseases has raised the possibility of eliminating senescent cells as an attractive therapeutic strategy. Several studies have shown that eliminating senescent cells with senolytics is associated with improvement of symptoms in animal models of various disorders [7,8,34]. Furthermore, there are several ongoing clinical trials to prove the use of senolytics in various fields [35].

The identification of senescence in tissues presents some difficulties, and currently, there is no single marker that identifies senescent cells; therefore, several characteristics need to be measured [16,36]. Some features of senescent cells, such as morphological changes, cell-cycle blockade and nuclear, mitochondrial and lysosomal changes, are used as biomarkers. SA-β-Gal, an indicator of activity of the acidic lysosomal β-galactosidase at pH 6.0, is the most widely used biomarker of senescence [29]. The induction of senescence by multiple stressors activates signaling pathways that converge on cell-cycle arrest. The increased expression of the cyclin-dependent kinase (CDK) inhibitor p16 (p16^INK4A^) is considered a suitable biomarker to determine stimuli-induced senescence in cells and tissues [37]. However, it is also expressed in certain non-senescent cells, such as macrophages [38], and it is not expressed by all senescent cells [39].

The staining for lipofuscin with the histochemical dye Sudan Black B is now considered a robust biomarker for senescence [28]. Lipofuscin is a lysosomal insoluble aggregate that accumulates during the aging of post-mitotic tissues or terminally differentiated cell types, such as neurons and cardiac myocytes, specifically good for identifying post-mitotic cellular senescence [40]. It has the advantage of being useful in paraffin-fixed samples.

Finally, using pro-inflammatory markers as members of SASP can be an essential step for identifying a senescent environment. The cytokines IL-6 and IL-8, which reinforce the senescence growth arrest, can be used as senescence biomarkers, although none of the individual SASP components are senescence specific [41]. Furthermore, there is heterogeneity in senescence related to the type of stressor, cell type, tissue and context in which it evolves [16]. Thus, since there are different approaches to detect senescent cells or tissues, the use of a single biomarker is not objective and can lead to erroneous interpretations. For all these reasons, in this study, several senescent markers were used.

Despite the interest in cellular senescence and the potential benefit of senolytics, there are few studies of senescence in AD. Our results in this study, showing an increased number of senescent cells in the frontal and hippocampal brain tissue, are in accordance with previous studies indicating a role for senescence in AD [42]. In the hippocampal slices, we found a trend of an increasing number of senescent cells as the severity of the disease increases, suggesting a continuum of accumulation of senescent cells in the brain as the disease progresses.

The presence of systemic manifestations of AD is nowadays well recognized [17,18,19,20,21,25,26,27,43]. In this study, we explored the presence of several senescent markers in PBMCs from aMCI and AD patients. As has been reported previously, the different markers of senescence were not homogeneously expressed, that is, some markers were increased, while others were unchanged or even reduced. Interestingly, the senescent markers in PBMCs appear to be differently expressed in aMCI and AD. We found that compared with HC, the PBMCs of aMCI patients showed an increase in several senescence markers, such as SA-β-Gal and G0/G1, as well as IL-6 and IL-8 mRNA expression and IL-8 plasmatic levels, suggesting that some markers of senescence might be increased as a peripheral manifestation at early stages of the neurodegenerative disease. Instead, the p53 expression showed a non-significant increase in the aMCI group, whereas the levels of γH2AX were comparable to controls. In the PBMCs of AD patients, on the other hand, SA-β-Gal, Go/G1 and IL-6 and IL-8 mRNA expressions—which were increased in aMCI PBMCs—had levels comparable to controls. Instead, the p16 and p53 expressions were significantly increased in AD PBMCs. This suggests that there might be a transient increase in some senescence markers at the initial stages of the disease that are no longer present at the more advanced stages.

Our results showed that γH2AX was significantly decreased in AD patients compared with healthy controls and aMCI. The results in the literature on γH2AX activation in the brain tissue are controversial. An increase in γH2AX activation evaluated by immunocytochemistry has been reported in astrocytes in the hippocampus of AD patients [44], suggesting that AD patients have selectively increased DNA damage. In contrast, another study [45] evaluated astrocyte DDR by γH2AX and DNA-dependent protein kinase expression in relation to AD Braak stages and found that neither marker increased with increasing Braak stage, suggesting a reduction in DNA repair in brains with increasing AD pathology. In peripheral tissues, increased levels of endogenous γH2AX have been reported in buccal cells and lymphocytes of MCI and AD patients compared with healthy controls [46,47]. However, our results indicate the opposite. A possible explanation for this variability is that DDR should be understood as a dynamic phenomenon, in which there is a balance between the generation of molecular damage and the activation of repair pathways. In this line, it has been observed that the activity of γH2AX can be transitory in time, in response to cytotoxic insults [48,49]. In this sense, our results, showing an increased expression of DDR pathways (p53, Figure 3) in the PBMCs of AD subjects, could be explained by the low levels of γH2AX activity (Figure 4).

Overall, the disparity of the different senescence markers we observed in PBMCs is in accordance with the high degree of heterogeneity described in the literature [16]. As mentioned before, it is interesting to point out that there might be a change in the capacity to induce senescence along with the progression of the disease; that is, PBMCs might lose some capacities with the progression of AD pathology. For example, the fact that PBMCs of AD patients do not express increased levels of IL-6 and IL-8 mRNA, as those of aMCI patients do, might imply that the capacity to release SASP factors decreases with disease progression, consistent with the change in the immune response that is observed along the course of AD [50]. Additionally, γH2AX, which is involved in DDR pathways and DNA repair, might decrease as the disease progresses. Finally, the drop in senescence markers seen in AD compared with aMCI PBMCs might also indicate a change in cell metabolism that favors cell death instead of senescence after a prolonged insult [40]. Under severe stress conditions, cells that can no longer undergo repair can either choose apoptosis or cell senescence [51]. It is known that the expression of cell-cycle-related proteins p16 and p53 is associated with senescence but also with apoptosis; p16 can induce the expression of apoptosis-related protein bcl-xL [52]. Our results, showing increased p16 and p53 expression in AD PBMCs, which does not correlate with other senescence biomarkers, might be explained by an increased selection of cell death instead of senescence at the advanced stages of AD. This is in accordance with our previous reports showing that lymphocytes from AD patients have an increased susceptibility to cell death after an oxidative insult [25,26]. Therefore, we propose that, at the initial stages of the disease, PBMCs manifest signs of senescence; instead, at later stages, they might undertake the death pathway. 

It is also interesting that the behavior of the markers in PBMC is not that of a progressive accumulation of senescent cells associated with disease progression, as occurs in the brain where severe AD patients show increased senescence mark, and less advanced cases seem to have a lower mark, but rather that PMBCs behave differently, with increases in several senescent markers at the early disease stages that later decrease as the disease progresses. This could be compatible with a dynamic response of PBMCs to the increased senescence developing in the brain tissue, a hypothesis that needs to be addressed with further experiments, in which a longitudinal study evaluating this phenomenon in PBMCs should be performed.

The reported changes in senescence markers in peripheral tissues from the early stages of the disease have multiple implications. On the one hand, it is relevant due to the need to develop biomarkers that facilitate the diagnosis of early stages of neurodegenerative diseases, and in this sense, our study helps propose new diagnostic objectives in this pathology. Nevertheless, these results are relevant, since they provide new information that supports the role of cellular senescence as a pathophysiological mechanism in AD and the possibility of studying senolytics as potential therapeutic options, in which changes in PBMCs senescence markers could constitute a possible biomarker to monitor treatment response.

Among the limitations of our study is that we worked with the entire PBMC population, so we do not know which leukocyte subpopulation is responsible. Our approach was aimed at searching for markers in the entire PBMC population because it is an easy procedure and has potential clinical applicability, unlike lymphocyte subpopulation sorting, which is not widely available. Another limitation is the small number of patients in certain conditions. Finally, our battery of markers could incorporate other reporters, such as p21. However, we believe our results provide good evidence to justify expanding this line of research in larger groups of patients and with other panels of potential markers.

In summary, this work reports the existence of senescence markers in PBMCs of patients at different stages of AD, and the results support further investigation to evaluate senolytics as a potential therapy for this pathology.

## 4. Materials and Methods

### 4.1. Subjects

Patients with aMCI, AD and healthy controls (HC) were recruited from Hospital Clínico Universidad de Chile. The study was approved by the Ethics Committee of Hospital Clínico Universidad de Chile, and all participants signed an informed consent form. Diagnoses of aMCI and AD were performed using the NIA-AA criteria [53]. Patients were evaluated by a neurologist and submitted to cognitive evaluation with the Clinical Dementia Rating (CDR) [54] scale and CDR sum of boxes (CDR-SOB) [55], the AD8 test [56], validated in Chile [57], and the Montreal Cognitive Assessment (MoCA) [58], validated in Chile [59]. The demographic characteristics of the study participants are shown in Table 1.

### 4.2. Neuropathological Samples

Frontal cortex sections from severe AD patients (CDR 3) and HC (CDR 0 at death) were kindly provided by the Alzheimer’s (ADRC) at the Washington University School of Medicine. Hippocampal brain sections of AD patients at Braak stages ranging from II to V and HC samples of similar age and sex were a gift from Dr Isidre Ferrer in Barcelona, Spain.

### 4.3. Staining of Lipofuscin

Sections of the frontal cortex or hippocampus (0.6 µm) of AD and HC were treated with the SenTraGor reagent, a staining method that measures the reaction of lipofuscin with the chemical compound (GL13) linked with biotin and with a second-step application of an enhancing immunohistochemical-enzymatic detection reaction.

### 4.4. Peripheral Blood Mononuclear Cells (PBMCs) Isolation

Peripheral blood was collected in sodium heparin vacutainer tubes. PBMCs were separated from whole blood by Ficoll-Hypaque TM PLUS density centrifugation, as previously described [25,26,60]. Cells were preserved in 1 ml TriZol Reagent (Ambion) and stored at −80 °C until mRNA extraction.

### 4.5. β-Galactosidase Activity

The β-galactosidase activity was determined using a fluorescence method described by [29]. To induce lysosomal alkalinization, 800,000 PBMCs were incubated with 100 nM bafilomycin A1 for 1 h in RMPI-1640 medium at 37 °C and 5% CO_2_. After that, 33 µM 5-dodecanoylaminofluorescein di-β–d-galactopyranoside (C_12_FDG), a fluorogenic substrate for β-galactosidase, was added and incubated for 2 h in the same conditions. The cells were washed 3 times with ice-cold phosphate-buffered saline (PBS) and re-suspended in PBS for the flow cytometer (FACSCanto, BD Biosciences). The mean fluorescence intensity (MFI) was used to estimate the β-galactosidase activity. 

### 4.6. G0-G1 Phase Cell-Cycle Arrest

The percentage of PBMCs arrested at the G0-G1 phase in the cell cycle was determined by flow cytometry with propidium iodide.

### 4.7. p16 and p53 Expression

Flow cytometry was performed from cryopreserved PBMCs. Cells were resuspended in a PBS solution supplemented with 10% fetal bovine serum. PBMCs were washed with PBS 3% fetal bovine serum (FBS) and fixed using the Cytofix-CytopermTM kit (BD Biosciences). The antibodies used were anti-human p16 (BD Pharmingen) and anti-human p53 (BD Pharmingen), and the secondary antibody was anti-rabbit Alexa 488 (Thermo Fisher Scientific, 1:1000). Cells were incubated with a specific antibody for 16 h at 4 °C, and then, the samples were washed twice with PBS 3% FBS. The secondary staining was performed in PBS 3% FBS for 1 h at room temperature and washed twice with PBS 3% FBS. Finally, the analysis was performed using a Cytoflex (Beckman Coulter) flow cytometer and FlowJo software v10.0.10. The cell population considered by each sample for the analysis was 100,000 cells.

### 4.8. RNA Isolation and PCR Analysis

Total RNA was isolated using the Trizol reagent. To remove any contaminating genomic DNA, a DNAse digestion step with TURBO DNA-freeTM Kit was included. cDNA was synthesized from total RNA (2 µg) using the High-Capacity cDNA Reverse Transcription Kit. Real-time quantitative PCR (qPCR) was performed in an amplification system MX3000P, using the DNA binding dye SYBR green (Brilliant III SYBER-GREEN Master Mix). Amplification was performed using the following primers: IL6: forward 5′-AACTCCTTCTCCACAAGCGCC-3′, reverse 5′-GTGGGGCGGCTACATCTTT-3′; IL8: forward, 5′-CTCTCTTGGCAGCCTTCCTGATT-3′, reverse 5′-AACTTCTCCACAACCCTCTGCAC-3; 18S: forward 5′-GATATGCTCATGTGGTGTTG-3′, reverse 5′-AATCTTCAGTCGCTCCCA-3′; SDHA: forward 5′-GAGGCAGGGTTTAATACAGCA-3′, reverse 5′-CCAGTTGTCCTCCTCCATGT-3′. All genes were normalized to the geometric mean of 18S and SDHA housekeeping genes. All samples were run in triplicate.

### 4.9. ELISA

The plasma levels of IL-6 and IL-8 were determined by ELISA, following the manufacturer’s recommendations. For IL-6, we used the Human IL-6 High Sensitivity ELISA Kit from eBioscience and for IL-8, the Human Ultrasensitive ELISA Kit from Thermo-Fisher.

### 4.10. γH2AX Immunocytochemistry

PBMCs were fixed with 1% formaldehyde fixation solution in phosphate-buffered saline (PBS) for 10 min at room temperature. Cells were rinsed three times with PBS and incubated with 0.1% Triton X-100 in PBS for 5 min and blocked with 2% BSA in PBS for 30 min. Cells were immunostained with anti-γH2AX (ser139) primary antibody (Merck) (1:3000, diluted in 0.2% BSA in PBS) at 4 °C overnight. After this incubation period, cells were rinsed three times with PBS and incubated with Alexa Fluor^®^ 488 anti-mouse secondary antibody (1:400, diluted in 0.2% BSA in PBS) for 1 h at room temperature. Cells were rinsed three times with PBS, and the coverslips were mounted in DAKO mounting medium on glass slides. 

Confocal image stacks were captured with a Nikon C2+ microscope, using a 20× objective. Between 100 and 120 cells per patient were acquired. ImageJ analysis software was used to generate the zeta projections from 10 to 15 stacks (0.8 mm thickness each). Quantitative analysis of immunofluorescence data was carried out with a histogram analysis of the fluorescence intensity at each pixel across the nucleus. The data were normalized by the number of nuclei and are expressed in arbitrary fluorescence units (AU).

### 4.11. Statistical Analysis

The statistical analysis used to compare the results between the groups was performed using Kruskal–Wallis with Dunn’s post hoc test for multiple comparisons. The data in each group were corrected for age and sex using multiple regression. All statistical analyses were performed using the GraphPad Prism 9 software. All values are expressed as the mean ± SEM. A *p*-value < 0.05 was considered statistically significant for all measurements.

## 5. Conclusions

This study shows that several senescence markers are present in the PBMCs of AD patients at different stages of the disease.

## Figures and Tables

**Figure 1 ijms-23-09387-f001:**
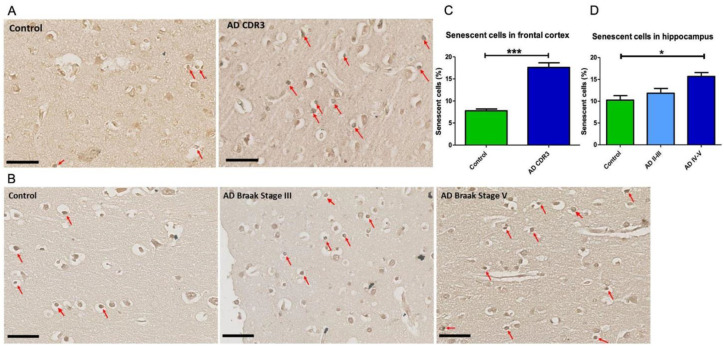
Senescent cells were increased in the hippocampus and frontal cortex of patients with advanced stages of AD. (**A**): Representative paraffin-embedded sections of frontal cortex of a healthy control (CDR 0, **left panel**) and an advanced AD patient (CDR3, Braak Stage VI, **right panel**) and (**B**) hippocampal sections of a healthy control (**left panel**), an AD Braak stage III patient (**middle panel**) and an AD patient (Braak stage V (**right panel**). Senescent cells indicated by arrows were identified with the lipofuscin reagent SenTraGor. Scale bars in (**A**,**B**): 50 μm. (**C**,**D**): Quantification of senescent cells in the frontal cortex (**C**) of five controls (CDR 0) and six severe AD patients (CDR3, Braak Stage VI) and the hippocampus (**D**) of three controls, three AD Braak stage II/III and four AD Braak stage IV/V patients. The number of senescent cells was increased in the frontal cortex and hippocampus of AD patients at advanced stages of the disease. Statistical analysis: t-test in C and one-way ANOVA with Bonferroni correction in D. * *p* < 0.05; *** *p* < 0.005.

**Figure 2 ijms-23-09387-f002:**
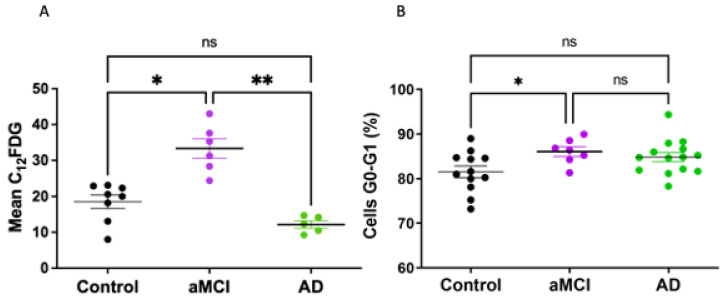
Senescent SA-β-Gal mark and growth arrest was increased in PBMCs of aMCI patients. (**A**) Senescent cells were identified by cytometric determination of SA-β-Gal activity using its fluorescent substrate C12FDG. SA-β-Gal activity was significantly increased in PBMCs of aMCI patients (n = 6) compared with healthy controls (n = 8), while AD PBMCs (n = 5) had levels similar to controls. (**B**) Growth arrest of PBMCs was determined by the percentage of cells at cell-cycle stage G0/G1 by cytometric determinations with propidium iodide staining. The percentage of PBMCs arrested at G0-G1 in aMCI patients (n = 7) was increased in comparison with healthy controls (n = 12), whereas no change was seen in AD PBMCs (n = 14). Statistical analysis: Kruskal–Wallis with Dunn’s post hoc correction, * *p* < 0.05, ** *p* < 0.01 for (**A**,**B**), ns = non-significant.

**Figure 3 ijms-23-09387-f003:**
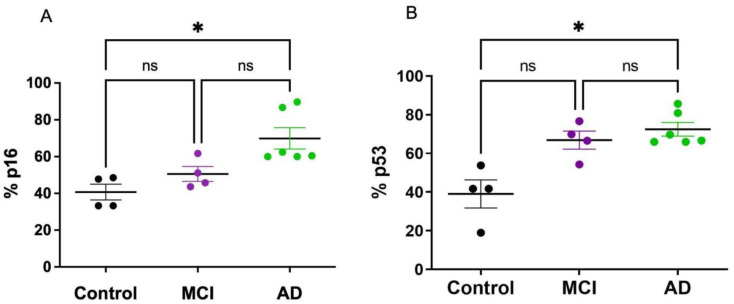
p16 and p53 expression was increased in PBMCs of AD patients. The expression of p16 (**A**) and p53 (**B**) was determined by flow cytometry in PBMCs from controls (n = 4), aMCI (n = 4) and AD (n = 6) patients. p16 and p53 expression was increased in AD patients compared with healthy controls, with no significant change in aMCI. Statistical analysis: Kruskal–Wallis with Dunn’s post hoc correction * *p* < 0.05, ns = non-significant.

**Figure 4 ijms-23-09387-f004:**
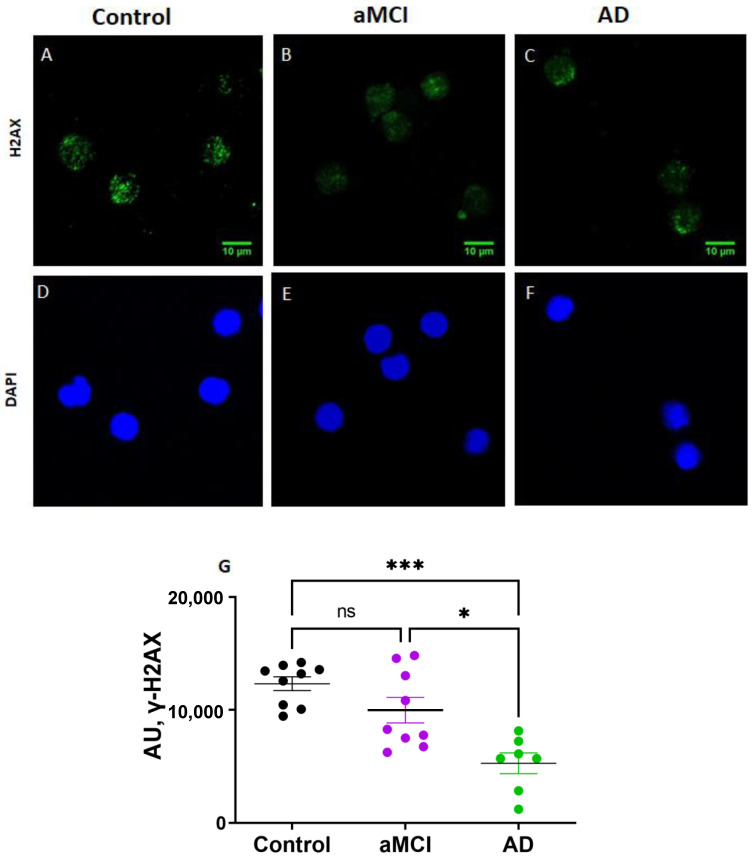
Activation of the DNA damage response determined by γH2AX activity was decreased in PBMCs of AD patients. (**A**–**F**): Representative immunofluorescence confocal images of γH2AX staining (green, (**A**–**C**)) and nuclei stained with DAPI (blue, (**D**–**F**)) of PBMCs of a control (**A**,**D**), an aMCI (**B**,**E**) and an AD (**C**,**F**) patient. Scale bars: 10 μm. (**G**): Immunofluorescence quantification of γH2AX expression normalized by the number of nuclei in nine healthy controls, nine aMCI and seven AD patients. AU: arbitrary fluorescence units. Statistical analysis: Kruskal–Wallis with Dunn’s post hoc correction, * *p* < 0.05, *** *p* < 0.005, ns = non-significant.

**Figure 5 ijms-23-09387-f005:**
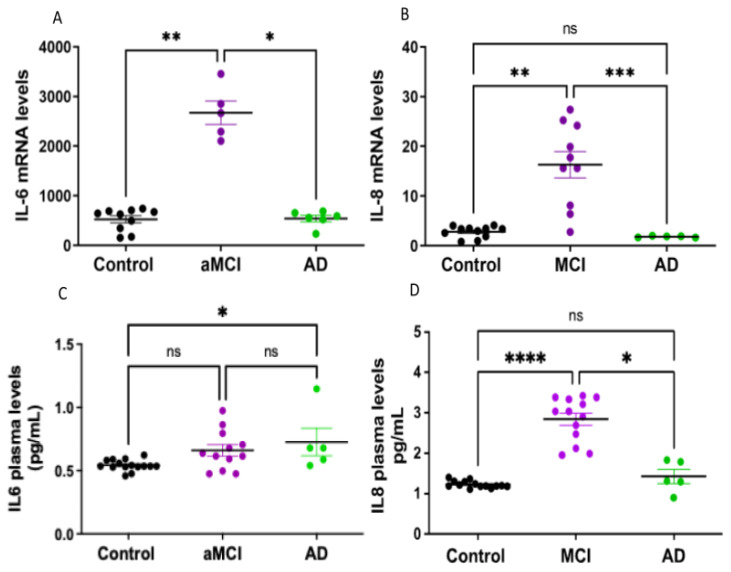
IL-6 and IL-8 expressions were increased in aMCI patients. (**A**,**B**): mRNA expression levels of cytokines IL-6 (**A**) and IL-8 (**B**) determined by qPCR in PBMCs of healthy controls, aMCI and AD patients. Both cytokines were increased in PBMCs of aMCI patients, whereas in AD, the levels were the same as in healthy controls. n= 19 healthy controls, 15 aMCI and 6 AD patients for IL6 and 11 healthy controls, 10 aMCI and 5 AD patients for IL8. (**C**,**D**): Plasmatic levels of IL-6 (**C**) and IL-8 (**D**) were measured by ELISA. IL-8 levels were significantly increased in aMCI compared with healthy controls and AD. There was a subtle increase in IL-6 plasmatic levels in AD, but the number of AD patients was small, and the values were too dispersed to be conclusive; n = 15 controls, 12 aMCI and 5 AD patients for IL-6 and 14 controls, 13 aMCI and 5 AD patients for IL-8. Statistical analysis: Kruskal–Wallis with Dunn’s post hoc correction. * *p* < 0.05, ** *p* < 0.01 *** *p* < 0.005, **** *p* < 0.0001, ns = non-significant.

**Table 1 ijms-23-09387-t001:** Demographic data of healthy controls, aMCI and AD patients in the study.

	Healthy Controls n = 26	aMCI n = 22	AD n = 12	*p*
Age, years, mean ± SE (range)	75.6 ± 1.8 (65–85)	77.2 ± 1.5 (63–93)	77.5 ± 1.9 (72–89)	=0.590
Female sex, (%)	17 (68.0)	16 (72.7)	11 (91.7)	=0.232
Education (years)	12.5 ± 1.0	10.1 ± 1.1	10.3 ± 1.5	=0.220
MoCA test score (mean ± SE)	28.2 ± 0.4	19.8 ± 0.9	11.8 ± 1.6	***p*****< 0.0001** (Ctr vs. aMCI and Ctr vs. AD); *p* = 0.065 (aMCI vs. AD)
AD8	0.4 ± 0.1	4.3 ± 0.5	6.4 ± 0.4	***p*****< 0.0001** (Ctr vs. aMCI and Ctr vs. AD); *p* = 0.186 (aMCI vs. AD)
CDR-SOB	0.1 ± 0.04	2.0 ± 0.2	7.1 ± 1.1	***p*****< 0.0001** (Ctr vs. aMCI and Ctr vs. AD); ***p* = 0.017** (aMCI vs. AD)
CDR 0 (number of patients)	26	0	0	***p*****< 0.0001** CDR0 vs. CDR > 0
CDR 0.5 (number of patients)	0	22	0	
CDR 1 (number of patients)	0	0	6	
CDR 2 (number of patients)	0	0	3	
CDR 3 (number of patients)	0	0	3	
Diabetes/Insulin Resistance, n (%)	6 (22.2)	5 (25.0)	3 (25.0)	=0.988
Hypertension, n (%)	19 (72.2)	9 (45.0)	4 (33.3)	**=0.025** (ctr vs. MCI and Ctr vs. AD)
Hypercholesterolemia, n (%)	7 (27.8)	9 (47.4)	6 (50.0)	=0.340

MoCA, Montreal Cognitive Assessment. AD8, Informant questionnaire Alzheimer’s Disease 8. CDR, Clinical dementia Rating. CDR-SOB, Clinical dementia Rating-Sum of Boxes. Chi Squared for categorical variables; Kruskal–Wallis with Dunn’s post hoc correction for numerical variables. Significant differences are marked in bold.

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
