# Peer review of "Senescence Markers in Peripheral Blood Mononuclear Cells in Amnestic Mild Cognitive Impairment and Alzheimer’s Disease"

_ijms, 2022, doi:10.3390/ijms23169387_

Round 1
Reviewer 1 Report
Major Comments:
1) The striking data in this manuscript is in Figure 3 which claims reduced levels of γH2AX in AD PBMCs compared with both controls and aMCI patients.
The image does not have a good magnification and the cells look tiny.
Also, the results are quite opposite, γH2AX activation is a marker for the induction of DNA damage and has been shown in several studies (https://doi.org/10.1186/s13024-020-00386-4). In other words, if the authors wanted to claim the presence of senescence and DNA damage in the AD PBMCs, they should have shown that γH2AX was increased in these cells compared to the control.
This need to be discussed with solid reasons and performing additional experiments such as western bloting, otherwise need to be removed from the manuscript.
2) Figure legends are not descriptive and are very short. More information (such as an explanation of the results) is required to make the readers have a better understanding of the figure
3) Figure 3 does not have any scale bar, and the size of the scale bar in figure 5 has not been mentioned.
Minor Comments:
1) Several references are missing:
Lines 46, 57, 79, 89
2) DDR has been defined several times in the manuscript (lines 29, 77, 105) also one of them is defined as “DNA-damage-repairing” which needs to be changed to DNA damage response.
3) AD has been defined more than one time. Page 1, page 2 (Twice),
4) PBMCs has defined more than one time on Page 1, Page 2, page 10
5) SASP has defined more than one time pages 1 and 2
6) SA-β-Gal has been defined more than once
7) γH2AX has appeared on the first page for the first time but has been defined in page 2
Author Response
We thank the reviewers for their valuable suggestions which we have addressed and added in the revised version of the manuscript. They have greatly improved our work. The changes in the revised version are marked in red to make it easier to visualize them. A detailed response to the reviewer’s comments follows.
1) The striking data in this manuscript is in Figure 3 which claims reduced levels of γH2AX in AD PBMCs compared with both controls and aMCI patients.
The image does not have a good magnification and the cells look tiny.
We have improved Figure 3 (Fig 4 in revised version) with better magnification of the PBMCs.
Also, the results are quite opposite, γH2AX activation is a marker for the induction of DNA damage and has been shown in several studies (https://doi.org/10.1186/s13024-020-00386-4). In other words, if the authors wanted to claim the presence of senescence and DNA damage in the AD PBMCs, they should have shown that γH2AX was increased in these cells compared to the control.
This need to be discussed with solid reasons and performing additional experiments such as western bloting, otherwise need to be removed from the manuscript.
We really appreciate the reviewer’s comments on this subject, and we have incorporated a discussion to address this issue. (The suggested reference: https://doi.org/10.1186/s13024-020-00386-4 must be a mistake since it refers to TDP43 in ALS). Our results show a decrease of γH2AX activation in AD with no change in aMCI patients. We agree with the reviewer that γH2AX is a marker for the induction of DNA damage and has been reported elevated in AD and other neurodegenerative diseases. Nevertheless, there is controversy in the literature about γH2AX activation in AD, both in brain and peripheral tissues, with increased and decreased levels of γH2AX, and another study indicating that the response of γH2AX to DNA damage does not correlate with AD severity. On the other hand, DDR is a dynamic phenomenon, in which there is a balance between the generation of molecular damage and the activation of repair pathways. In this line, it has been observed that the activity of γH2AX can be transitory in time, in response to cytotoxic insults (Mariotti et al 2013, Bernadotte et al 2016). Our results showing that PBMC from AD subjects have increased expression of DDR pathways (p53, Figure 3 revised version) could explain the low levels found in γH2AX activity (Figure 4 revised version). We consider that our results are contributing to the discussion on the response to DNA damage in peripheral tissue in AD; and that, like what has already been described in brain tissue, this is an open field that requires further evaluation. We added a paragraph in the Discussion section considering this issue.
Mariotti, L.G.; Pirovano, G.; Savage, K.I.; Ghita, M.; Ottolenghi, A.; Prise, K.M.; Schettino, G. Use of the γ-H2AX Assay to Investigate DNA Repair Dynamics Following Multiple Radiation Exposures. PLoS One 2013, 8, e79541, doi:10.1371/JOURNAL.PONE.0079541.
Bernadotte, A.; Mikhelson, V.M.; Spivak, I.M. Markers of Cellular Senescence. Telomere Shortening as a Marker of Cellular Senescence. Aging (Albany. NY). 2016, 8, 3–11, doi:10.18632/AGING.100871.
Our results of H2AX protein expression were made with immunofluorescence technique with a highly specific antibody for phospho-Histone H2AX (Ser139), which allows determining nuclear location and protein expression. Immunofluorescence is a reliable and sensitive method for quantitatively measuring DNA double-strand breaks (DSBs) and is widely used to determine H2AX expression. Even though we agree that western blot experiments would complement our results, we believe the immunofluorescent data are indicative as a first step.
2) Figure legends are not descriptive and are very short. More information (such as an explanation of the results) is required to make the readers have a better understanding of the figure.
We thank the reviewer for the comment and we have incorporated an explanation in all figure legends.
3) Figure 3 does not have any scale bar, and the size of the scale bar in figure 5 has not been mentioned.
Thank you for noticing our mistake. We added the scale bar in fig 3 (now Fig 4). The size of the scale in Fig 5 (now Fig 1) is indicated.
Minor Comments:
1) Several references are missing:
Lines 46, 57, 79, 89
We added the references in lines 46, 57, 79 and 89 of the original version, as were solicited.
2) DDR has been defined several times in the manuscript (lines 29, 77, 105) also one of them is defined as “DNA-damage-repairing” which needs to be changed to DNA damage response.
3) AD has been defined more than one time. Page 1, page 2 (Twice),
4) PBMCs has defined more than one time on Page 1, Page 2, page 10
5) SASP has defined more than one time pages 1 and 2
6) SA-β-Gal has been defined more than once
7) γH2AX has appeared on the first page for the first time but has been defined in page 2
Thank you for pointing out this. We corrected the abbreviations and mistake.
Reviewer 2 Report
The manuscript has some aspects of interest, but the manuscript needs to be restructured, the numbers of subjects are low, the elisa dosages are very low, the results published in this year which are reported below have not been considered and must be considered and discussed. Some considerations on the various PMNC components (lymphocytes (T cells, B cells & NK cells), monocytes and dendritic cells).
1-Cao M, Liu J, Zhang X, Yang T, Wang Y, Hou Y, Song Q, Cui Y, Wang Y, Wang P. ABI3 Is a Novel Early Biomarker of Alzheimer's Disease. J Alzheimers Dis. 2022;87(1):335-344. doi: 10.3233/JAD-215635. PMID: 35275543.
2-Wang X, Wang D, Su F, Li C, Chen M. Immune abnormalities and differential gene expression in the hippocampus and peripheral blood of patients with Alzheimer's disease. Ann Transl Med. 2022 Jan;10(2):29. doi:10.21037/atm-21-4974. PMID: 35282083; PMCID: PMC8848377.
3-Garfias S, Tamaya Domínguez B, Toledo Rojas A, Arroyo M, Rodríguez U, Boll C, Sosa AL, Sciutto E, Adalid-Peralta L, Martinez López Y, Fragoso G, Fleury A. Peripheral blood lymphocyte phenotypes in Alzheimer and Parkinson's diseases. Neurologia (Engl Ed). 2022 Mar;37(2):110-121. doi: 10.1016/j.nrleng.2018.10.022. Epub 2021 Feb 12. PMID: 35279225.
Author Response
We thank the reviewers for their valuable suggestions which we have addressed and added in the revised version of the manuscript. They have greatly improved our work. The changes in the revised version are marked in red to make it easier to visualize them. A detailed response to the reviewer's comments follows:
Reviewer 2 comments:
the manuscript has some aspects of interest, but the manuscript needs to be restructured, the numbers of subjects are low, the elisa dosages are very low, the results published in this year which are reported below have not been considered and must be considered and discussed. Some considerations on the various PMNC components (lymphocytes (T cells, B cells & NK cells), monocytes and dendritic cells).
1-Cao M, Liu J, Zhang X, Yang T, Wang Y, Hou Y, Song Q, Cui Y, Wang Y, Wang P. ABI3 Is a Novel Early Biomarker of Alzheimer's Disease. J Alzheimers Dis. 2022;87(1):335-344. doi: 10.3233/JAD-215635. PMID: 35275543.
2-Wang X, Wang D, Su F, Li C, Chen M. Immune abnormalities and differential gene expression in the hippocampus and peripheral blood of patients with Alzheimer's disease. Ann Transl Med. 2022 Jan;10(2):29. doi:10.21037/atm-21-4974. PMID: 35282083; PMCID: PMC8848377.
3-Garfias S, Tamaya Domínguez B, Toledo Rojas A, Arroyo M, Rodríguez U, Boll C, Sosa AL, Sciutto E, Adalid-Peralta L, Martinez López Y, Fragoso G, Fleury A. Peripheral blood lymphocyte phenotypes in Alzheimer and Parkinson's diseases. Neurologia (Engl Ed). 2022 Mar;37(2):110-121. doi: 10.1016/j.nrleng.2018.10.022. Epub 2021 Feb 12. PMID: 35279225.
We really appreciate the reviewer’s comments that have improved our manuscript.
We restructured the manuscript as solicited incorporating the reviewer's suggestions. We inverted the order of the results, starting with the brain tissue experiments (this is not marked in red).
Regarding the comment about the number of subjects, we included 60 patients in all (26 healthy controls, 22 aMCI and 12 AD patients), which is a significant number, although we agree that in some measurements the number of patients was small. However, most of the results have an important number of patients and we were able to find significant results. We added a paragraph in the discussion section mentions this as a limitation of our study.
Regarding low Elisa dosages (we assume it is Elisa concentrations). We found very variable values of IL6 and IL8 in plasma and we agree that the reported blood levels are higher. However, we have treated all the samples in the same way, and we have focused on the comparison between groups.
Thank you for indicating the recent reports that appeared in the literature this year in relation to peripheral manifestations of AD. We have incorporated them in the new version of the manuscript.
Round 2
Reviewer 2 Report
The observations and comments were welcomed by the authors
Round 3
Reviewer 2 Report
The authors have addressed and revised the manuscript according to reviewer's comments and suggestions.